# Calculation of a Reference Interval for Rectal Temperature in Adult Dogs Presenting for Veterinary Care Using an Algorithm for Mixed Data

**DOI:** 10.3390/ani14131970

**Published:** 2024-07-03

**Authors:** Elisabeth Dorn, Kirsten Bogedale, Alexander Pankraz, Reto Neiger

**Affiliations:** 1Small Animal Department, Faculty of Veterinary Medicine, Ghent University, 9000 Ghent, Belgium; 2Clinic of Small Animal Medicine, LMU University of Munich, 80539 Munich, Germany; 3Biocontrol, Veterinary Division of Bioscientia Healthcare GmbH, 55218 Ingelheim, Germany; alexander.pankraz@biocontrol.de; 4IVC Evidensia DACH, 80336 Munich, Germany; reto.neiger@ivcevidensia.de

**Keywords:** body temperature, canine, data warehouse, fever, reference range

## Abstract

**Simple Summary:**

The body temperature of dogs is an integral part of clinical examinations in which veterinarians evaluate a dog’s health. Surprisingly, there are no studies based on many dogs defining a reference rectal temperature range. This study aimed to determine a reference interval for rectal temperature in adult dogs using data from a large number of healthy and diseased dogs and to test an algorithm for calculating these ranges. In total, 24,013 temperature records from 9782 adult dogs were obtained from the health records of dogs seen at a university clinic between 2008 and 2017. Repeated measurements, records without age, dogs younger than one year old, and extreme temperatures (below 30.0 °C and above 43.0 °C) were excluded. Using a specialized algorithm, 665 outliers were identified, and 9117 measurements underwent further statistical analysis. Based on these, the mean rectal temperature was found to be 38.6 °C, with a reference interval of 37.7–39.5 °C. This method confirmed existing reference intervals and can help establish new ones.

**Abstract:**

Veterinarians rely on the measurement of canine body temperature to define the health status of dogs, but no studies exist defining a reference range for rectal temperature on a large group of dogs. The aim of this study was to define the rectal body temperature of dogs based on a large data set of diseased and healthy animals and to evaluate the capability of the employed algorithm to calculate reference intervals of numerical clinical data. Out of 24,013 recorded measurements, statistical analysis was applied to data from 9782 adult dogs that underwent clinical examination at a university clinic between 2008 and 2017. The reference interval was calculated using an algorithm developed by the Deutsche Gesellschaft für Klinische Chemie und Laboratoriumsmedizin e.V. as part of its Reference Limit Estimator software (version 1.40.36.07). The following values were excluded: multiple measurements in a given dog, samples without assigned age or dogs younger than one year, and values <30.0 °C and >43.0 °C. Out of 9782 adult dogs, 665 temperature measurements were identified as outliers, and 9117 were used for further statistical analysis. The mean rectal temperature was 38.6 °C (90% CI: 38.6–38.6 °C) with a reference interval of 37.7 °C (90% CI: 37.7–37.7 °C) to 39.5 °C (90% CI: 39.5–39.5 °C). Validation according to CLSI guidelines showed the results to be valid. The determination of a reference interval for rectal temperatures in dogs using an algorithm for mixed datasets yielded results comparable to the existing reference intervals. This demonstrates that the calculation of reference intervals from mixed datasets of clinical numerical data can be used to confirm existing reference intervals or establish such de novo.

## 1. Introduction

Body temperature is a commonly measured parameter to define the health status of dogs. Increased temperature can be due to fever or hyperthermia, the former seen in inflammation, neoplasia, or immune mediated disease, the latter in increased muscle activity, nervousness or overheating [1]. Low temperature may indicate shock, cardiac disease or exposure to cold temperatures and is a frequent complication of anesthesia [2]. Body temperature is also used to define health status prior to vaccination or elective surgeries [1]. In dogs, body temperature is most commonly measured by rectal thermometry whereas other methods, such as auricular, axillary [3] or non-contact thermometry [4] have also been published. Rectal temperatures in healthy dogs have been reported to range from 37.9 °C to 39.9 °C (100.2–103.8 °F) [5], but it is unclear how this reference interval was established [1,2,6].

Reference intervals in veterinary medicine are commonly established based on a relatively small population of healthy, young animals. Accordingly, these are calculated by a nonparametric method of percentile estimates with confidence intervals to determine the 95th percentile interval (2.5th–97.5th percentile range) for results from >120 clinically healthy animals [7].

Consequently, studies on the rectal temperature of dogs are typically conducted on a small number of predominantly healthy animals (Table 1).

A different approach of defining reference intervals is data mining to obtain reference data from existing databases. The concept of population-based reference intervals was introduced in human medicine in 1969 and subsequently applied to veterinary species [8,9]. The American Society for Veterinary Clinical Pathology (ASVCP) has recommended adherence to Clinical and Laboratory Standards Institute (CLSI) guidelines for the determination of reference intervals including the use of data mining to obtain reference intervals from existing data [7]. This approach does not follow the common theory of reference values, which is the careful definition of the reference population [10]. Therefore, this system is burdened with the uncertainty of the current health status of the individual. The first publications were based on the statement that the majority of laboratory results are “normal”. Therefore, considering the distribution of results, statistical procedures were applied to eliminate the extremes of the distribution curve, leading to exclusion of the less frequent results, typical of ‘unhealthy’ subjects [11].

Considering the amount of data from healthy individuals together with the development of tools to separate pathologic from healthy individuals, this approach is promising [12]. This new approach has recently been published for the establishment of clinical chemistry reference intervals with a statistical analysis of an existing dataset obtained from a large database [13]. To the author’s knowledge, there are currently no published studies in veterinary medicine that investigate clinical parameters of physical examination using large datasets.

Studies based on the unknown health status of patients have shown to be appropriate for the analysis of large datasets [14]. Furthermore, they are useful for the establishment of reference values in a large number of patients, where data are based on patients with an unknown health status [15]. Furthermore, when the health status of the individuals in the analyzed sample is unknown, these indirect methods for estimating reference intervals from large datasets are considered a consistent approach [16].

In this study, a reference interval for the rectal temperature of a large cohort of clinically healthy and diseased adult dogs is calculated using an existing dataset with the Reference Limit Estimator algorithm of the Deutsche Gesellschaft für Klinische Chemie und Laboratoriumsmedizin e.V. (DGKL), under the assumption that most dogs have a normal temperature during clinical examination.

**Table 1 animals-14-01970-t001:** Previous literature on the rectal temperature of dogs.

Study	Number of Dogs	Health Status	Rectal Temperature	Aim of the Study
Veronesi et al., 2002 [17]	7	healthy, pregnant	at the onset of whelping: 37.6 ± 0.7 °C. After 12 h 38.3 ± 0.9 °C, 24 and 36 h after the onset of whelping (38.7 ± 0.3 and 38.5 ± 0.2 °C).	Correlation of temperature, progesterone, cortisol and prostaglandin of the periparturient bitch
Refinetti and Piccione, 2003 [18]	7	healthy	mean 39.1 °C, SD 0.01 °C	Daily rhythmicity of body temperature
Sousa et al., 2011 [19]	88	healthy	median 38.8 °C, SD 0.4 °C	Comparison between auricular and rectal temperature measurements
Lamb and McBrearty, 2013 [3]	212	unknown	median 38.0 °C, mean 37.9 °C, range 33.9–40.4 °C	Comparison of rectal, auricular and axillary temperature measurements
Goic et al., 2014 [20]	94	unknown	median 38.9 °C, range 36–40.8 °C	Comparison of rectal and axillary temperature measurements
Gomart et al., 2014 [21]	250	hospitalized	median 38.0 °C, range 35.0–40.4 °C, SD 0.85 °C	Comparison of rectal, auricular and axillary temperature measurement and associated stress response in hospitalized dogs
Konietschke et al., 2014 [22]	238	healthy and diseased	mean 38.1 °C, reference range 37.2–39.2 °C, 95% CI 38.0–38.3 °C	Comparison of auricular and rectal temperature measurement in normothermic, hypothermic and hyperthermic dogs
Osinchuk et al., 2014 [23]	12	healthy	mean 38.7 °C, range 37.6–39.5 °C, SD 0.37 °C	Comparison of ingestible temperature sensor and rectal temperature measurements
Kreissl and Neiger, 2015 [24]	300	healthy and diseased	median 38.3 °C, range 35.5–41.1 °C, 95% CI 38.2–38.4 °C	Comparison of ocular and rectal temperature measurement
Yanmaz et al., 2015 [25]	30	healthy	mean 38.0 °C, 95% CI, lower bound (°C): 37.8, upper bound (°C): 38.2, SD 0.12 °C	Comparison of rectal, ocular and auricular temperature measurements
Zanghi, 2016 [26]	32	healthy	median 38.0 °C ± 0.5 °C at rest, 39.7 °C ± 0.9 °C during exercise	Comparison of ocular, auricular and rectal temperature measurement at rest or with exercise
Cichocki et al., 2017 [27]	50	healthy	mean 38.2 °C (SD 0.88 °C)	Comparison of axillary, auricular and rectal temperature measurement in healthy dogs
Hall and Carter, 2017 [28]	24	healthy	median 38.3 °C, mean 38.3 °C, range 37.4–39.1 °C, SD 0.39 °C	Comparison of rectal and auricular temperature measurement in healthy exercising dogs
Cugmas et al., 2020 [29]	204	unknown	mean 38.1 °C, SD 1.0 °C;	Comparison between rectal and body surface temperature in dogs

## 2. Materials and Methods

### 2.1. Selection Criteria

In this retrospective, single-center study, the rectal temperature data of dogs presented to a university small animal clinic in Germany between January 2008 and June 2016 was used. Dogs were not excluded based on clinical presentation, health status or diagnosis. Furthermore, exclusion was not based on factors such as time of day, seasonality, food intake or estrus cycle in female dogs. All dogs underwent a complete clinical examination, including obtaining rectal temperature, irrespective of the presenting complaint. This was also the case in dogs presented for health checks. Rectal measurements were performed by students and staff; all participants were instructed on how to measure the rectal temperature (length and depth). Measurements were taken with various digital predictive thermometers without the use of lubricant. The rectal thermometers were not calibrated. The thermometer was inserted into the rectum for a length of about 2 cm. The measurement of a single rectal temperature took 10–20 s and was immediately recorded in the electronic patient management system.

Recordings of clinical examination results including breed, weight, age and sex were entered into individual fields in the patient management system, making it possible to export these parameters individually from the database. If the temperature of an animal was measured multiple times, the temperature determined upon initial presentation was used. The following measurement results were excluded: dogs with an unclear age, dogs younger than one year, and values of <30.0 °C (<86.0 °F) or >43.0 °C (>109.4 °F) due to being considered errors in documentation. Initially, 24,013 rectal body temperature measurements from 12,051 dogs were collected. After applying exclusion criteria, data from 9782 adult dogs were included in the statistical analysis. Exclusions were due to age less than 1 year (2214 dogs), unspecified age (48 dogs), duplicate measurements (11,962 measurements), or temperatures outside the expected range of <30 °C or >43 °C (7 dogs). The subsequent analysis was conducted using a single measurement from each of the 9782 dogs.

### 2.2. Statistical Analysis

The reference interval for canine body temperature was calculated using an algorithm developed by the Deutsche Gesellschaft für Klinische Chemie und Laboratoriumsmedizin e.V. (DGKL) as part of its Reference Limit Estimator software (https://www.dgkl.de (accessed on 21 November 2019)), which was incorporated into the Abacus add-in for Excel. This approach assumed that, in a mixed data set of healthy and diseased subjects, the central part of the population (i.e., the highest peak of a histogram) represented the non-pathological values with high probability. Outliers in the dataset were identified according to Tukey as being outside of the 1.5 interquartile range (IQR) of the first and third quartile [14]. The interquartile range was defined as the range of values that are in the middle 50% between the lower and upper quartile (25th and 75th percentile). The data set, after truncation by the 1.5 IQR rule and DGKL algorithm, was tested for a normal distribution using the Anderson–Darling test and was furthermore visually inspected for a normal distribution using the Q-Q-plot.

The expected distribution of temperature was considered as that within an upper value (1.5*IQR+upper quartile) and a lower value (1.5*IQR-lower quartile). An unexpected temperature signified an outlier, and was defined as a data point that was outside the expected temperature distribution of the population [30]. This was carried out because the DGKL algorithm is susceptible to bias when a large number of outliers is present [16]. This approach optimized the Kolmogorov–Smirnov distance between the cumulative densities of the model and the data [31] using a smoothened kernel density function. The calculation method requires a dataset of >1000 data points. The abovementioned method has successfully been used to calculate reference limits for various laboratory parameters [31,32,33].

Validation of the reference interval was performed according the CLSI guidelines and 20 temperature values of apparently healthy dogs, excluding outliers, were used and compared to a known reference interval. The known reference interval was obtained from the international Merck veterinary manual being between 37.5 °C and 39.2 °C. Reference intervals for an analyte are considered valid if 0–2 results fall inside the proposed reference interval. If more than 4 results fall outside the proposed reference limits, the reference interval should be rejected and de novo reference intervals determined. When 3 or 4 results fall outside the proposed reference interval, an additional 20 samples can be collected and interpreted as described previously [7].

## 3. Results

The temperature data of 9782 adult dogs were subjected to statistical analysis. A total of 244 different breeds, crossbreeds (2596 dogs), and 45 dogs without a noted breed were enrolled in the study. The median age was 6.8 years with a range of 1.0–23.7 years. Dogs were female intact (*n* = 2281), female spayed (*n* = 2479), male intact (*n* = 2883), male neutered (*n* = 2124), and 15 without noted gender. Outliers were removed based on the working guidelines by the DGKL Reference Limit Estimator software. Overall, 665 measured rectal temperatures were identified as outliers by the 1.5 IQR rule, and 9117 measurements were used for further statistical analysis. The distribution of temperatures is visualized in Figure 1. 

The dataset before the removal of outliers was found to be nonparametric as determined by the Anderson–Darling test (*p*-Values < 0.0001). The truncated dataset used by the algorithm for the reference interval calculation was found to be normally distributed by visual inspection of the Q-Q plot (Figure 2).

For the final 9177 measurements of dogs in the dataset, the mean rectal temperature was 38.6 °C (101.5 °F) (90% CI: 38.6–38.6 °C) and the lower reference value was 37.7 °C (99.9 °F) (90% CI: 37.7–37.7 °C), whereas the upper reference value was 39.5 °C (103.1 °F) (90% CI: 39.5–39.5 °C) (Figure 3). Validation according to the CLSI guidelines showed that a single temperature value was outside the calculated reference interval confirming the new reference interval.

## 4. Discussion

This is the first study defining a reference interval for the rectal temperature in dogs using a mixed data set. Veterinary textbooks and published reports variously offer normal reference intervals of 37.5–39.9 °C (99.5–103.8 °F) [5,34,35] which is compatible with the finding of our study (37.7–39.5 °C).

Furthermore, this study has shown the potential of using mixed datasets to calculate novel reference intervals; however, further testing is needed.

The common process of defining a reference interval is the “direct approach”, meaning analyses are performed on selected subjects representing the reference population [36]. A different approach is the “indirect” approach, which has been used in the present study. Based on this, results were collected from subjects as part of routine pathology testing prior to the usage of statistical techniques to define reference intervals [37]. Since these datasets include healthy and unhealthy subjects, further steps were described to reduce the likelihood of disease affecting the results. Therefore, it was recommended to limit results to a single result per patient, since a diseased patient is more likely to be retested [38]. Accordingly, only one measurement per patient was included in the present study. Furthermore, the exclusion of more extreme results in a dataset, which are the ones likely to be affected by disease was described [38]. Accordingly, calculation of the data without the removal of the outliers in the present study yielded results that were not comparable to reference intervals for body temperature from the literature. Considering the statistical approach, it has been confirmed in this study that the algorithm is susceptible to bias when a large number of outliers is present [16,39]. However, after removal of the outliers, the reference interval was compatible with previous definitions and was successfully validated according to CLSI guidelines.

Pollution of the dataset with many pathological values can lead to erroneous results, as shown with the calculation using the DGKL algorithm without excluding outliers. However, the method is designed to be used on a data set that includes pathological values. After the elimination of outliers by the 1.5 IQR test, the DGKL algorithm yields reference intervals that are comparable with those from the literature, which also show some variation depending on the source. The DGKL algorithm is capable of detecting a reference population in a mixed data set of pathological and non-pathological values. An abundance of pathological values can cause bias in the calculation; thus, outliers were removed.

Several factors are considered to have an impact on the rectal temperature of healthy dogs. It is well known that excitement, stress, food intake, or exercise can increase the body temperature [40]. Furthermore, the body temperature is influenced by age with newborns not having the ability to maintain a constant body temperature [41] and young animals having higher temperatures compared to adults [34]. Therefore, only dogs older than one year were included in the present study. No further testing for animals younger than one year was performed, since this subgroup was too heterogenous (e.g., the young animals should be subdivided by age in months).

It was reported that hormones have an impact on the thermogenesis of female dogs [42]. A decrease in progesterone results in a decrease in body temperature [42]. The body temperature may help to detect the stages of estrus cycle and ovulation time in dogs [43]. A physiological change in body temperature is also seen in dogs during and after birth [17]. Higher temperatures during the first days after parturition are considered to be normal [17]. Accordingly, the body temperature of bitches after parturition is slightly elevated [44]. In the present study, it is unknown at what time in the estrus cycle the temperature of female dogs was taken. Due to the high number of intact females (*n* = 2281), the hormonal effects most likely will have no effect on temperature measurements, as dogs will present during various times of the estrus cycle. Furthermore, no statistical difference could be seen in rectal temperature between intact and spayed female dogs. (*p* < 0.001).

Some information can be gained from reports in which rectal temperature was taken for other reasons than determining reference intervals (Table 1). The study populations in these investigations consisted of healthy dogs [25,28,45,46], dogs in certain conditions such as pregnancy [17], or dogs with and without otitis externa [47]. Other studies were based on dogs that were presented to a veterinary clinic, including sick ones [22,24], dogs with an unknown health status [3,20], hospitalized dogs [21], systemically healthy ones undergoing elective surgery [27] and dogs of any condition undergoing surgery [29]. Some studies on rectal temperature specifically excluded dogs with rectal disease [20,21,27]. In the present study, dogs were included regardless of their health status.

In contrast to the recent studies on the rectal temperature of dogs, the present study is the first to define a reference interval for the rectal body temperature by examining a large, mixed population of healthy and diseased animals (9177 dogs). Based on the CLSI guidelines, which outline standard methods of creating reference intervals [36], calculation of a reference interval should be performed by direct methods using data of a healthy reference population ideally including at least 120 individuals. It is a frequently used approach that the reference interval should include the central 0.95 fraction (or 95%) of the data of the reference population [10]. The indirect approach with the DGKL Reference Limit Estimator utilizes a mixed data set, which in our case comprises data from 9177 dogs. The mean temperature was determined to be 38.6 °C (101.5 °F), with a reference interval of 37.7–39.5 °C (99.9–103.1 °F). Our study confirms that a reference interval for canine rectal body temperature can be calculated using a large mixed dataset. The size of the final reference population (9177 dogs—more than 80 times the minimum CLSI requirement for reference populations) remained substantial despite rigorous screening and exclusion criteria [36].

### Study Limitations

Measuring the rectal temperature can contain several sources of error. A study in cows showed variations up to 0.5 °C (0.9 °F) due to the measurement procedure, the type of thermometer (increase by 0.3 °C or 0.5 °F, respectively) and the penetration depth into the rectum (increase by 0.4 °C or 0.7 °F, respectively) [48]. For the correct measurement of the rectal temperature the thermometer has to be inserted into the rectum for at least 2 cm [49]. Changes in the rectal measurement are recognized slower than they really happen [50]. Erroneously high measurements can happen by measuring too early after exercise or food intake [51]. Local changes, like inflammation or hyperemia of the rectum can lead to false high values. Erroneously low measurements can be caused by obstipation or insufficient sphincter tonus [52]. Tonicity and peristalsis can also have an impact on temperature measurement [53]. The accuracy of digital thermometers is limited by their displays. Commonly, there is only one decimal figure. However, all these limitations are seen in everyday practice during clinical examination and taking rectal temperature.

The dogs were examined at the clinic at various times of the day, some during stressful circumstances, but this also represents the everyday situation of animals being presented to a veterinary clinic. Furthermore, it is possible that our findings are impacted by geographical locations, climatic conditions, or veterinary clinic practices. The rectal temperatures in the present study were measured once during clinical examination by staff and students in their clinical rotation and not by one single examiner. The temperature was recorded directly after obtaining the result, therefore false data entry is deemed unlikely. Since taking rectal temperature is part of the clinical education of students, it is assumed that students took the temperature correctly. Different digital predictive thermometers were used in this study, the thermometers were not calibrated. These facts illustrate the routine practices in veterinary practices, where different people use various, uncalibrated, thermometers to measure temperature during a single clinical examination. Our data, collected from dogs presented under diverse conditions and circumstances, accurately reflect the typical procedures of a veterinary practice. The results show that the reference temperature interval of dogs presented to a veterinarian might be broader than previously thought, implying that in a dog, a rectal temperature of 39.5 °C in a hospital environment can be normal. However, this mildly wider reference interval in rectal body temperature must now be further validated in the clinical setting.

## 5. Conclusions

The mean rectal temperature in dogs was determined to be 38.6 °C (101.5 °F), with a reference interval of 37.7–39.5 °C (99.9–103.1 °F). Our study confirms that a reference interval for canine rectal body temperature comparable to previously published reference intervals can be calculated using a large mixed dataset. The size of the final reference population (9177 dogs—more than 80 times the minimum CLSI requirement for reference populations) remained substantial despite rigorous screening and exclusion criteria.

## Figures and Tables

**Figure 1 animals-14-01970-f001:**
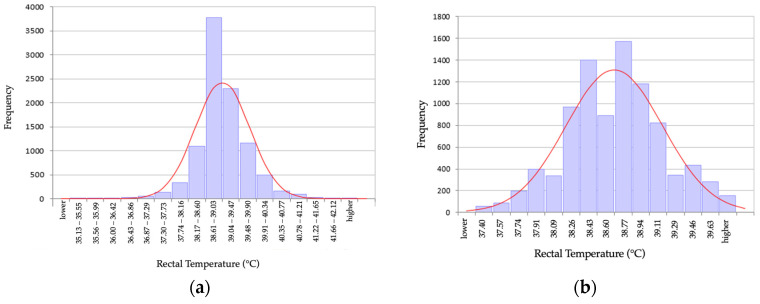
Histogram of the distribution of rectal temperature in (**a**) 9782 adult dogs before removal of outliers; and (**b**) in 9117 adult dogs after removal of 655 outliers.

**Figure 2 animals-14-01970-f002:**
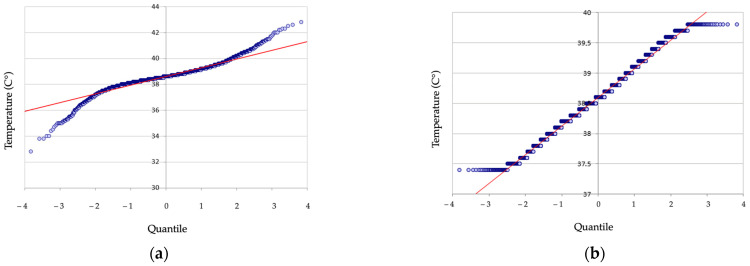
Q–Q plot of rectal temperature data (**a**) of 9782 adult dogs before removal of outliers showing non-linear distribution and (**b**) in 9117 dogs after removal of 655 outlier data points now showing a linear distribution.

**Figure 3 animals-14-01970-f003:**
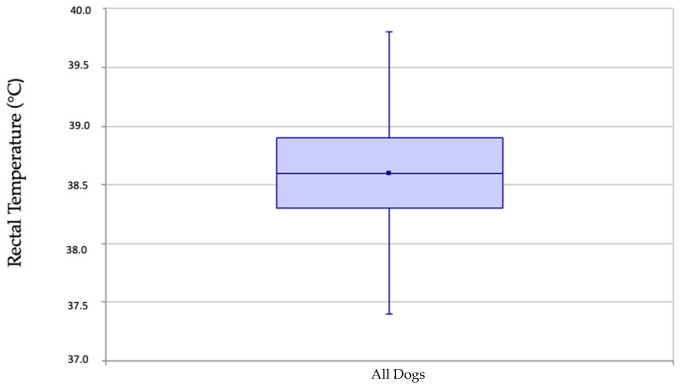
Visualization of the rectal temperature of 9117 adults dogs via box and whisker plot. The upper and lower boundary lines are defined by the 75% and 25% quantiles of the measurement results. Within this box, the median is marked by a horizontal line and the mean by a square point. The vertical lines (“whiskers”) represent the range of the sample, with the endpoints of the lines defined by the largest and smallest data points still within 1.5 times the interquartile range.

## Data Availability

The raw data supporting the conclusions of this article will be made available by the authors on request.

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
