# Peer review of "Calculation of a Reference Interval for Rectal Temperature in Adult Dogs Presenting for Veterinary Care Using an Algorithm for Mixed Data"

_animals, 2024, doi:10.3390/ani14131970_

Round 1

Reviewer 1 Report

Comments and Suggestions for Authors

Precise assessment of dog temperature, measured by the traditional rectal method, is a basic diagnostic tool for every veterinarian. However, the measurement result obtained must be verifiable and must indicate the health condition of the animal, and more specifically of a specific individual (assessment in terms of age, sex, breed or body weight);

The authors of the manuscript collected a great deal of data, which were properly systematized and presented. The research methods and the concept of the work are described in detail, however, I would like to point out a few comments for future correction.

In the Chapter results [124], the authors included the stage of data preparation, i.e. admission and elimination of subjects from the study. Preparing a dataset is not data analysis, as the authors erroneously point out. This stage should be described in the chapter Research material, the result of which is the final indication of the number of n individuals whose temperature measurement results were given statistical analysis. The description of the results seems to be chaotic and incomprehensible, its graphic presentation is poor. The authors' assumption about a mixed data set, which, when subjected to the indicated algorithm, will allow to obtain reference values for temperature measurement, is interesting and well-thought-out, but presented in an unattractive and disordered way. The presented results should correspond to the aim of the work, and their summary in conclusions should correspond to the set goal. In the case of the manuscript, we have 2 basic sets: healthy and sick dogs and the usefulness of the algorithm to indicate the range of reference values. This is how the results should be sorted and described, from the general to the specific, which may still show further positive or negative responses to the authors' goal of the study.

The lack of charts, which are a perfect form of presentation of results in this type of manuscript, were omitted by the authors for incomprehensible reasons.

The database collected by the authors is so extensive that it is really worth working on presenting the results of an indisputably important research topic.

After introducing corrections and re-review, I believe that the work can be very good and useful in the practical aspect.

Reviewer 2 Report

Comments and Suggestions for Authors

This article is a study on the reference range of normal rectal temperature in dogs, which uses a mixed dataset algorithm to calculate.

Ensure that the dataset used in the study represents a wide range of dog breeds, ages, and health conditions to improve the universal applicability of the reference interval. Clearly indicate the rectal temperature measurement techniques used in the study, including the type of equipment used, insertion depth, and measurement time, to reduce the impact of operational differences on the results. Further refine the exclusion criteria, such as excluding dogs that have engaged in vigorous exercise or food intake for a short period of time before measurement, as these factors may affect body temperature readings. Visualization of results: Provide charts to display the distribution of data. In the discussion section, discuss in more detail the significance of these findings for veterinary practice and how they can be applied to clinical diagnosis.

Provide explanations of the the following probable inadequacies.

The study used data from a single center, which may result in sample selection bias, which may affect the universal applicability of the results. The study mentioned the use of different digital thermometers for measurement, but did not provide detailed information on whether these devices have been calibrated or whether the measurer has received standardized training, which may affect the accuracy of the measurement. The study did not provide a detailed explanation of the environmental conditions during measurement, such as temperature, humidity, etc., which may affect the dog's body temperature. Although the study mentioned gender and sterilization status, it did not provide a detailed explanation of how these factors affect body temperature measurement. The research results may be limited by specific geographical locations, climatic conditions, or veterinary clinic practices.

Reviewer 3 Report

Comments and Suggestions for Authors

Thank you for sharing this manuscript, you have identified a true gap in the current canine temperature literature, and have presented a novel study aiming to address this gap using an impressively large dataset. While your methods appear robust, there are a number of fundamental considerations that should be reviewed, and potentially addressed to ensure the reporting is not misleading. Firstly, there is no descriptive data for the dataset used in this analysis, with no apparent testing for the need for data partitioning. The addition of a histogram of the raw data, and further details on the outliers would is required as a minimum. The second key concern, is the inclusion of dogs with multiple readings, despite recommendations these animals should be excluded on the basis of being more likely to be “unhealthy” due to the need for repeated testing. Ideally, the reference interval would be re-calculated without these animals, to test the effect of including them. If this can not be performed (for logistical reasons) then this must be addressed in the limitation section. Finally, any de-novo reference interval should be validated and tested before being used clinically, as per the CSLI guidelines [7]. Again, if it is not possible to add this to the current study, this must be acknowledged in the limitations and conclusions. The claim that this reference interval is “consistent” is not currently supported by the data presented, please define how you judge “consistency”, or rephrase the sections that make this claim.

Specific points for consideration:

Title: Calculation of a reference interval for normal rectal temperature in dogs using an algorithm for mixed data.

-        This title is potentially misleading, whilst I appreciate the “normal” is in reference to the central distribution of temperatures forming the reference range, as your dataset includes non-healthy animals it would perhaps be more appropriate to be more descriptive in the title, e.g., “Calculation of a reference interval for rectal temperature in dogs presenting for veterinary care using an algorithm for mixed data”? Throughout the rest of the manuscript the interval is referred to simply as “reference interval for rectal temperature in dogs” with no mention of normal, so it is misleading to add it for the title alone.

Simple summary – consider further simplifying some of the language in this section to make this more accessible to non-expert readers?

Abstract – it seems odd not to include the reference interval (and confidence intervals) here in the abstract? A statement qualifying the provisional nature of this interval (without further validation) should be added.

Introduction:

Table 1 – please correct the study citations in column one, currently only the first authors are listed whereas many citations require the second author or et al. adding for accuracy. There are also a number of errors in this summary of the published work, please check specifically:

Gomart et al., (2014) compared rectal, auricular and axillary temperatures.

Konietschke et al., (2014) reported a “reference range” of 37.2-39.2C not just a “range” (albeit without reporting the methods of establishing that reference range), and the confidence interval reported was 38.0–38.3C (not 38.1-38.3C) in relation to the mean (as currently presented it appears to relate to the reference interval).

Hall & Carter (2017) compared rectal to auricular temperature, not ocular.

Line 60 – whilst I agree that using statistics to reduce the number of less frequent “unhealthy” results is a sound justification for using big data to explore RIs for laboratory data, where multiple parameters are screened with every sample when only 1 or 2 parameters may be of primary interest to the clinician ordering the test, the same is perhaps not true of physiological parameters? How often do the clinicians in this hospital record “temperature normal” in lieu of a true measurement? Are healthy dogs’ temperatures routinely checked during routine appointments, or is it perhaps likely that temperature is more likely to be checked if the clinician has a concern about the dog’s temperature/health status? And what proportion of consultations at the hospital are for “healthy” animals undergoing routine preventative health care, versus requiring clinical evaluation? Is there not the potential for this dataset to be skewed towards the unhealthy population? Can you quantify the likely number of healthy individuals included in the dataset?

Line 78 – please define DGKL at first use.

Methods:

Line 95 – What is the justification for setting >43C as being “considered errors in documentation”? Shapiro et al (1973) documented rectal temperatures up to 44.6C in their experimental heatstroke work, and numerous heatstroke studies have documented temperatures >43C in clinical patients, with healthy exercising dogs reaching 42.5C in several studies. >43C is therefore potentially an accurate body temperature measurement, unless, the thermometry devices being used do not record beyond 43C, which would be a different rationale to the one presented.

Line 106 – acknowledging this limitation, and potential for bias, would it be prudent to include a critical threshold limit for automatically excluding temperatures? Lewis and Foster (1976) and more recently Bruchim et al., (2017) have suggested >41C as the critical threshold beyond which pathological changes of heatstroke are possible in dogs. Ramsey and Tasker [1] report that temperatures beyond 41.1C are rarely observed in true fever, adding further weight to using 41C as a cut off for inclusion? This would reduce the number of pathological samples in the dataset and reduce the likelihood of bias.

Validation? Friedrichs et al. [7] recommend validating any de-novo reference intervals, using additional reference individuals to test the interval proposed, and also suggest evaluating the subsequent 90% confidence intervals of the limits as a test for the reliability of the interval. Was any validation performed on this interval?

Results:

Given the volume of data available, would it not be prudent to perhaps explore the need for interval partitioning? Across other species, and indeed with several canine studies, the variables age, sex and bodyweight have been proposed as factors influencing body temperature, was any evaluation performed on this data? To simply present the reference interval, with no detail on the underlying dataset (minimum, maximum, interquartile range) the reader can not truly evaluate the results being presented. A histogram of the dataset used to establish the reference interval should be included as a minimum, which would allow the reader to visualise the data distribution, and ideally where the outliers were located (primarily at the top or bottom of the dataset range).

As the confidence limits are incredibly narrow, this could be used to evaluate the range as recommended by Friedrichs et al. [7].

Discussion:

Line 149 – please add a supporting citation.

Line 158 to 162 – the statement that “only one measurement per patient was included” does not follow the argument presented in the preceding sentence, that “a diseased patient is more likely to be retested”. If a diseased patient is more likely to be re-tested, then arguably, dogs with multiple readings should have been excluded? Admittedly, if the readings are a year apart (e.g. subsequent annual vaccination appointments) then these animals should be retained, but for dogs with multiple readings during a single visit/stay, or dogs with multiple readings suggesting repeat visits within a short time frame, this logic would support excluding those dogs, not simply using the first reading?

Line 162 – this statement further supports the need to present a more comprehensive overview of the overall dataset within the results.

Line 173 – at the moment, your claims about the validity of the proposed reference interval is based on comparing it to previously published intervals, that you identified as lacking robust empirical evidence in the introduction. Including a validation of your interval would strengthen this argument.

Line 190 – consider expanding on this argument. As you included dogs from multiple years (presumably from multiple seasons?), even if bitches in different stages of estrus were included, due to the sample size any variation would be accounted for in the data?

Line 206 – check phrasing, missing “include”?

Line 211 – what do you mean by “consistent reference interval”? Do you mean an interval that is comparable to previously published sources?

Limitations – please include some discussion on the points listed above, specifically the lack of a validation test, the inclusion of individuals with multiple readings (despite acknowledging these animals are more likely to be diseased), and the proportion of preventative examinations versus pathological examinations in the dataset.

Future perspectives – please consider removing this section, as it currently repeats arguments presented in the main discussion with no additional information.

Conclusion:

Line 257 – consider expanding the first sentence for clarity “The median rectal temperature for dogs was…”.

Line 258 – again, the statement “consistent reference interval” is not supported by the data presented. If multiple reference intervals were calculated using difference datasets analysed in this manner, then consistency could be claimed, but at most the interval presented can be described as “comparable to previously published reference intervals”.

A major limitation as yet not acknowledge is the need to validate this reference interval. Given the novel application of using a mixed dataset to establish a physiological parameter reference interval (rather than tested clinical pathology, laboratory based intervals) there is a need to test the robustness and accuracy of this reference interval before recommending application to clinical practice.

Please confirm that this study received ethical approval?

Round 2

Reviewer 1 Report

Comments and Suggestions for Authors

I accept all the corrections made and recommend the manuscript in its current form for printing.

Author Response

Thank your for your help and guidance to improve the quality of our paper.

Reviewer 2 Report

Comments and Suggestions for Authors

 I agree that this manuscript could be accepted in the current form.

Author Response

(The authors gave the same response as above.)

Reviewer 3 Report

Comments and Suggestions for Authors

Thank you for responding to my original comments so carefully and comprehensively, I hope you'll agree the manuscript is now a stronger representation of your considerable work. 

I have a few minor comments to consider on the revised version, where clarity could be improved or phrasing addressed:

Line 119 – do you mean that 11,962 dogs were excluded due to duplication of measurement here, or do you mean 11,962 measurements were excluded? I suspect the latter.

Line 147 – should this read healthy dogs (plural)? 20 additional results from a single dog would not meet the validation criteria.

Figure 1 – this may be my personal preference, but could these two histograms be presented with the same X/Y axis values to allow direct comparison?

Figure 2 – please check the axis labels for consistency.

Figure 3 – please label the X axis.

Line 200 – ideally include the overall range and median of the 20 validation temperatures collected so the reader can appreciate how these results compare to the calculated reference interval.

Line 219 – This is a rather bold claim, given the validation presented includes just 20 additional samples and a comparison to previously published reference intervals (that did not cite their source), and given your acknowledgement that the interval requires further validation in a clinical setting in the limitation section. As Friedrichs et al. outline more comprehensive methods for testing and validating novel reference intervals (such as collecting sufficient additional “healthy” reference samples to calculate an additional de novo reference interval for comparison), I’m not sure you can claim to have fully validated the reference interval presented here. Please rephrase this claim to something like “this study has shown the potential of using mixed datasets to calculate novel reference intervals, however further testing is needed”.

Line 262 – either “entire females” or “non-neutered females” would be more typical phrasing as castration is usually only applied to the act of removing the gonads of males? Also, introducing data in the discussion should be avoided, why not add this analysis to the methods/results as it would further strengthen your study!

Line 313 – this is a really important point and acknowledges the real-world application of this study and the findings, and the need to collect data that reflects the setting it will eventually be used in! Great addition.

Comments on the Quality of English Language

There are a few really minor phrasing points that could be improved as highlighted in the points above.

Author Response

Reviewer: 3
Comments:

Thank you for responding to my original comments so carefully and comprehensively, I hope you'll agree the manuscript is now a stronger representation of your considerable work.

        We thank the reviewer again for dedicating their time and expertise to evaluate our manuscript. We appreciate the thorough reviewing of the manuscript and are convinced that taking into account the comments of the reviewers will enhance the quality of the paper.

I have a few minor comments to consider on the revised version, where clarity could be improved or phrasing addressed:

Line 119 – do you mean that 11,962 dogs were excluded due to duplication of measurement here, or do you mean 11,962 measurements were excluded? I suspect the latter.

        Indeed, 11,962 measurements were excluded.

We rephrased the sentence and hope it is less confusing now: “Initially, 24,013 rectal body temperature measurements from 12,051 dogs were collected. After applying exclusion criteria, data from 9,782 adult dogs were included in the statistical analysis. Exclusions were due to age less than 1 year (2,214 dogs), unspecified age (48 dogs), duplicate measurements (11,962 measurements), or temperatures outside the expected range of <30°C or >43°C (7 dogs). The subsequent analysis was conducted using a single measurement from each of the 9,782 dogs.“

Line 147 – should this read healthy dogs (plural)? 20 additional results from a single dog would not meet the validation criteria.

        Thank you for this comment. Indeed, it was “dogs” and accordingly changed in the manuscript.

Figure 1 – this may be my personal preference, but could these two histograms be presented with the same X/Y axis values to allow direct comparison?

        The partitions of the histograms are automatically generated by the Abacus software. Consequently, we are unable to provide the reader with two histograms that have identical partitions on the X-axis.

Figure 2 – please check the axis labels for consistency.

        Thank you for this comment. The graphs were adjusted.

Figure 3 – please label the X axis.

        Thank you for this comment. It was added.

Line 200 – ideally include the overall range and median of the 20 validation temperatures collected so the reader can appreciate how these results compare to the calculated reference interval.

        Due to technical reasons, we are unable to report the overall range and median of the 20 validation temperatures.

Line 219 – This is a rather bold claim, given the validation presented includes just 20 additional samples and a comparison to previously published reference intervals (that did not cite their source), and given your acknowledgement that the interval requires further validation in a clinical setting in the limitation section. As Friedrichs et al. outline more comprehensive methods for testing and validating novel reference intervals (such as collecting sufficient additional “healthy” reference samples to calculate an additional de novo reference interval for comparison), I’m not sure you can claim to have fully validated the reference interval presented here. Please rephrase this claim to something like “this study has shown the potential of using mixed datasets to calculate novel reference intervals, however further testing is needed”.

        Thank you for this suggestion. We rephrased it to the following: “Furthermore, this study has shown the potential of using mixed datasets to calculate novel reference intervals, however further testing is needed.”

Line 262 – either “entire females” or “non-neutered females” would be more typical phrasing as castration is usually only applied to the act of removing the gonads of males? Also, introducing data in the discussion should be avoided, why not add this analysis to the methods/results as it would further strengthen your study!

        Thank you for taking this into consideration.

We rephrased the sentence in “discussion” to: “Furthermore, no statistical difference could be seen in rectal temperature between entire and spayed female dogs.”

We would prefer not to put this finding into “results”. We feel that adding these results to the results section would automatically mean that we should also add more data such as difference between age groups, between breeds, between gender etc - and this is exactly what was never the plan. So, leaving only this information without adding it to results does not give too much emphasis to this information.

Line 313 – this is a really important point and acknowledges the real-world application of this study and the findings, and the need to collect data that reflects the setting it will eventually be used in! Great addition.

        Thank you!

Comments on the Quality of English Language

There are a few really minor phrasing points that could be improved as highlighted in the points above.

        Thank you again for your helpful suggestions.
